# Characterizing Spatiotemporal Patterns of Winter Wheat Phenology from 1981 to 2016 in North China by Improving Phenology Estimation

**Shuai Wang** [1,2], **Jin Chen** [2], **Miaogen Shen** [2], **Tingting Shi** [3], **Licong Liu** [2], **Luyun Zhang** [2], **Qi Dong** [2] **and Cong Wang** [4,*]

1  Department of Surveying and Mapping Engineering, Minjiang University, Fuzhou 350108, China
2  State Key Laboratory of Earth Surface Processes and Resource Ecology, Faculty of Geographical Science, Beijing Normal University, Beijing 100875, China
3  College of Economics and Management, Minjiang University, Fuzhou 350108, China
4  Key Laboratory of Agricultural Remote Sensing (AGRIRS), Ministry of Agriculture and Rural Affairs, Institute of Agricultural Resources and Regional Planning, Chinese Academy of Agricultural Sciences, Beijing 100081, China
*  Correspondence: wangcong01@caas.cn

**Abstract:** Phenology provides important information for wheat growth management and the estimation of wheat yield and quality. The relative threshold method has been widely used to retrieve phenological metrics from remotely sensed data owing to its simplicity. However, the thresholds vary substantially among phenological metrics and locations, hampering us from effectively detecting spatial and temporal variations in winter wheat phenology. In this study, we developed a calibrated relative threshold method based on ground phenological observations. Compared with the traditional relative threshold method, our method can minimize the bias and uncertainty caused by unreasonable thresholds in determining phenological dates. On this basis, seven key phenological dates and three growth periods of winter wheat were estimated from long-term series (1981–2016) of the remotely sensed Normalized Difference Vegetation Index for North China (106°18′–122°41′E, 28°59′–39°57′N). Results show that the pre-wintering phenological dates of winter wheat (i.e., emergence and tillering) occurred in December in the south and in mid- to late- October in the north, while the post-wintering phenological dates (i.e., green-up onset, jointing, heading, milky stage, and maturity) exhibited the opposite pattern, that is, January to May in the south and February to June in the north. Consequently, the vegetative growth period increased from 49 days in the south to 77 in the north, and the reproductive growth period decreased from 51 days to 29 days. At the regional scale, all winter wheat phenological dates predominantly advanced, with the most pronounced advancement being for green-up onset (–0.10 days/year, $p > 0.1$), emergence (–0.09 days/year, $p > 0.1$), and jointing (–0.08 days/year, $p > 0.1$). The vegetative growth period and reproductive growth period at the regional scale predominantly extended by 0.03 ($p > 0.1$) and 0.09 ($p < 0.001$) days/year, respectively. In general, the later phenological events (i.e., heading, milky stage, and maturity) tended to advance with higher temperature, while the earlier phenological events (i.e., emergence, tillering, green-up onset, and jointing) showed a weak correlation with temperature, suggesting that the earlier events might be mainly affected by management while later ones were more responsive to warming. These findings provide a critical reference for improving winter wheat management under the ongoing climate warming.

**Keywords:** climate change; phenology; threshold method; vegetation index; winter wheat

## 1. Introduction

Northern China accounts for 72.2% of the total area and 80.1% of wheat cultivation in the country. Winter wheat production in this region is important for ensuring domestic

food security and has a significant impact on the global wheat trade [1,2]. This region has experienced substantial warming since the early 1980s, and the warming has accelerated since the early 2010s [3]. The increasing temperature can modify the growth process of crops and the phenological dates (or length of the growth period), which will affect the yield and quality of crops [4–8] and have important implications for food security. In such a context, information about the responses of winter wheat phenology to temperature is important for understanding the response of winter wheat to climate change and for adaptive planting and improved cultivation management practices [9].

Current studies on winter wheat phenology are mainly based on ground phenology observation [10–13] and remote sensing extraction [2,14,15]. Ground-based observations record different phenological dates of crops by manual observations at individual agro-meteorological stations according to predefined criteria [16,17]. The advantage of the ground observation method is that the observed phenology is more clearly defined, and can better correspond to the different growth periods of winter wheat. It also has the advantages of high accuracy and high frequency. However, it cannot provide temporally and spatially continuous phenological information in a large area because of its time-consuming and labor-intensive nature [18]. The large spatial variability of wheat phenology [19] and the uneven spatial distribution of limited agro-meteorological stations lead to difficulty in completely characterizing long-term wheat phenology across a continuous space [15,20,21].

Remote sensing provides an alternative tool to investigate the spatiotemporal dynamics of phenology at a regional or global scale [22]. Currently, three types of methods have been applied in detecting phenology from vegetation index (VI) time series, namely, the curve fitting method, curve matching method, and relative threshold method [18]. The curve fitting method first fits the VI time series with a predetermined mathematical function (e.g., logistic function), and then it determines the phenological date by the feature points on the fitted curve (e.g., maximum values of the derivative, minimum or maximum values in the rate of change in curvature) [1,23,24]. The curve matching method first defines a shape model or reference curve with given phenological dates, and then it matches the target vegetation index curve to the predefined curve (i.e., shape model) by model fitting, such as shape model fitting (SMF) [25] or cross-correlation [26]. The relative threshold method defines a phenological date when the VI reaches a predetermined value (i.e., threshold) [27]. The relative threshold method has been widely used because of its simplicity and ease of implementation. Taking the green-up onset date, for example, the commonly used relative thresholds are 10%, 20%, and 50% [25,28,29]. However, these thresholds are mostly determined empirically and there is currently no suitable method to determine the relative thresholds for different phenological dates. Even for the same phenological date, the relative threshold could vary spatially over large areas, leading to uncertainty and bias in extracting phenological dates with a fixed relative threshold. Therefore, there is an urgent need to develop a method to determine the relative thresholds of multiple phenological dates (including emergence, tillering, green-up onset, jointing, heading, milky stage, and maturity) in different areas to improve the accuracy of phenology estimation.

Although a large number of studies have been conducted on the spatiotemporal characteristics of wheat phenology and their drivers based on remotely sensed data, most of these studies have been limited to a few specific phenological dates, such as the green-up onset date or heading date because of the lack of methods to estimate multiple phenological dates of winter wheat from satellite data [30]. However, the lack of methods hampers the quantification of spatial variations and temporal changes of the other phenological dates (i.e., emergence, tillering, jointing, milky stage, and maturity) and how these phenological dates respond to climate warming. This further leads to a deficiency in the assessment of the difference in the changes and temperature responses among various phenological dates as well as the lengths of the vegetative and reproductive growth periods.

Therefore, we need to develop a unified method to detect various phenological dates of winter wheat and their spatial and temporal changes. To address this, we first developed a method to determine the relative thresholds of different phenological dates in different

locations based on ground phenological observations to estimate multiple phenological dates from normalized difference vegetation index (NDVI) time series, then systematically studied the spatiotemporal characteristics of multiple winter wheat phenological dates and the lengths of the vegetative and reproductive growth periods, and investigated the relationship between each of the phenological dates (or lengths of growth periods) and temperatures.

## 2. Materials and Methods

### 2.1. Study Area

The study area is the main winter wheat-producing area in China (Figure 1), including the Huanghuai and the Jianghuai wheat-growing region (106°18′–122°41′E, 28°59′–39°57′N). This region covers eight provinces (Hebei, Henan, Shanxi, Shandong, Shaanxi, Jiangsu, Hubei, and Anhui) and has the most favorable natural environment (e.g., warm climate and moderate rainfall) for wheat growth in China [31]. Winter wheat in this region is usually sown in October and harvested around mid-June of the following year [32]. This region contributes significantly to China's total winter wheat production [1,2].

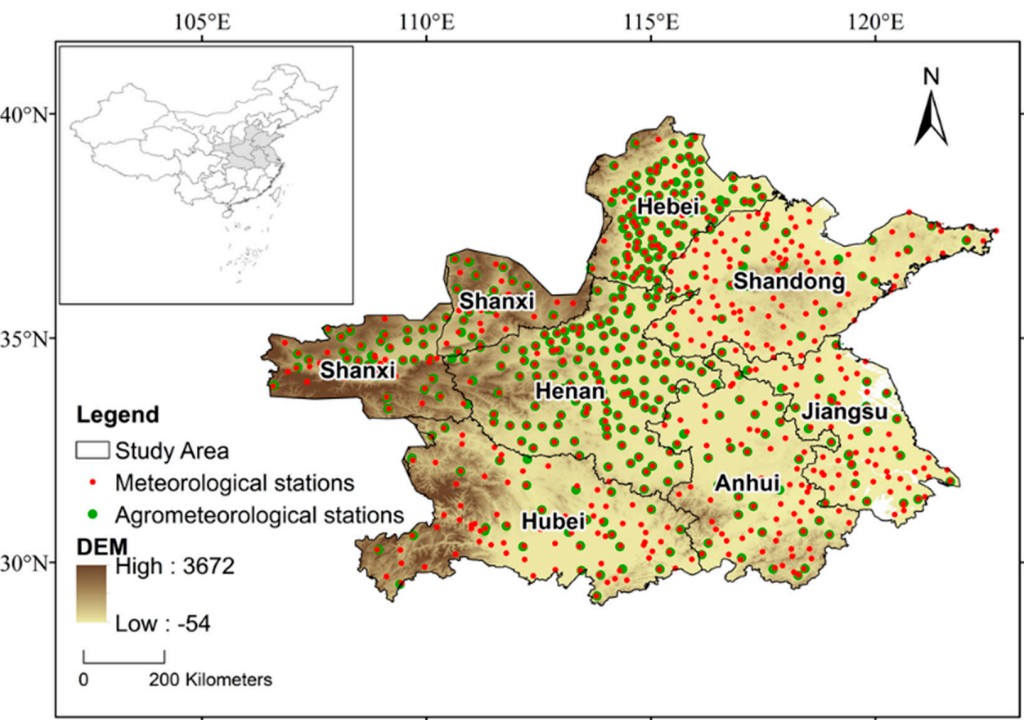

**Figure 1.** Study area and locations of meteorological and agrometeorological stations.

### 2.2. Datasets

#### 2.2.1. Remotely Sensed Data

The winter wheat phenology from 1981 to 2016 was estimated from NDVI time series from the NASA Making Earth System Data Records for Use in Research Environments (MEaSUREs) Vegetation Index and Phenology (VIP) collection. The VIP collections provide more than 30 years of a consistent global record for vegetation indices and landscape phenology based on Advanced Very High Resolution Radiometer (AVHRR) and Moderate Resolution Imaging Spectroradiometer (MODIS)/Terra MOD09 surface reflectance data. The 7-day composite NDVI product with 0.05 degree spatial resolution in VIP collections (VIP07) was downloaded from the LP DAAC website (https://lpdaac.usgs.gov/products/vip07v004/, accessed on 1 October 2022). The NDVI as well as quality assurance/pixel reliability of the VIP07 product were used to generate NDVI time series from early September to the end of July of the following year, covering the complete wheat-growing cycle. To remove noise

caused by cloud contamination and poor atmospheric conditions, NDVI time series were smoothed by the Savitzky–Golay (SG) filter [33–36]. The smoothed NDVI time series were then used for phenology extraction.

In addition, the Global Multi-resolution Terrain Elevation Data (GMTED2010) were downloaded from the U.S. Geological Survey (USGS) website (https://earthexplorer.usgs.gov/, accessed on 1 October 2022) to provide topographic data (i.e., altitude) of the study area. Latitude, longitude, and altitude extracted from the topographic data were then used to estimate the phenological date of each wheat pixel (more details follow in Section 2.3.1).

### 2.2.2. Wheat Classification Maps

It is unlikely that the winter wheat planting area remained the same for more than 30 years during the study period (1981–2016). Therefore, wheat classification maps for each year were needed to identify wheat pixels. The annual wheat classification map was extracted based on the Combining variations Before and After estimated Heading dates (CBAH) algorithm [37]. This algorithm distinguishes wheat pixels by designing two indices based on the differences and variable ranges of VI time series during the early and late growth stages of winter wheat. Pixels with large values of these two indices are considered to be the wheat pixels. Thresholds were set as 0.3 and 0.12, respectively, as suggested by Qiu et al [37].

### 2.2.3. Ground Observation Data

The ground observations included daily meteorological data and ground-observed phenology records. Daily meteorological data from 1981 to 2016 were collected from the China Meteorological Administration (CMA), which were subjected to strict quality control with a correct data rate of close to 100%. The ground-observed phenology records for 1993–2016 were obtained from the China Meteorological Data Sharing Service System (http://data.cma.cn/, accessed on 6 May 2021). The key phenological metrics of winter wheat in the agro-meteorological station, including the dates of sowing, emergence, tillering, dormancy, green-up onset, jointing, heading, milky stage, and maturity, were recorded. By linear regression against altitude, latitude, and longitude [37,38], each of these dates was spatially interpolated to each wheat pixel in the study area, which was then used to determine the optimal relative threshold for the phenological date.

### 2.3. Methods

We first developed the Calibrated Relative Threshold Method (CRTM) based on ground phenology observations and applied it to the estimation of the winter wheat phenological dates, including emergence, tillering, green-up onset, jointing, heading, milky stage, and maturity. On this basis, the lengths of different growth periods were calculated, including the Vegetative Growth Period (VGP), Reproductive Growth Period (RGP), and the total length of these two growth periods (Vegetative and Reproductive Growth Period, VRGP). According to recent studies, VGP is defined as the period from green-up onset to heading [20], RGP is defined as the period from heading to maturity [7,15], and VRGP is defined as the growth period from green-up onset to the maturity of winter wheat. Finally, linear regression was used to calculate the temporal trends of the seven phenological dates and three growth periods. The relationship between each of the phenological dates and the mean preseason temperature (defined in Section 2.3.3) and relationships between each growth period length and temperature were investigated by simple linear correlation analysis. We did not include precipitation in the correlation analysis because there was an effective irrigation system in this region.

### 2.3.1. The Calibrated Relative Threshold Method

We developed the calibrated relative threshold method (CRTM), which uses the ground-observed phenological date to calibrate the relative threshold of the corresponding phenological date (Figure 2). The core of the method is to determine the relative thresholds

of phenological dates from the multiyear average NDVI time series using ground-based phenological observations. More specifically, for each agro-meteorological station, the multiyear average phenological date was calculated from its ground phenological records of more than 20 years (i.e., 1993–2016). Due to the relatively sparse distribution of the agrometeorological stations, the ground-based observations could not cover the entire spatial extent of the study area. A recent study has shown that the key winter wheat phenological date is mainly controlled by the cumulative temperature, which is well correlated with latitude, longitude, and elevation [37–39]. Therefore, the phenological date of each wheat pixel was estimated by regressing the multiyear average phenological date of all stations across the study area (i.e., 378 stations) against the station's latitude, longitude, and elevation (Table 1) based on topographic data (i.e., GMTED2010). By applying the multiyear average phenological date to the multiyear average cumulative NDVI curve, the relative threshold for the phenological date of the specific wheat pixel was calculated.

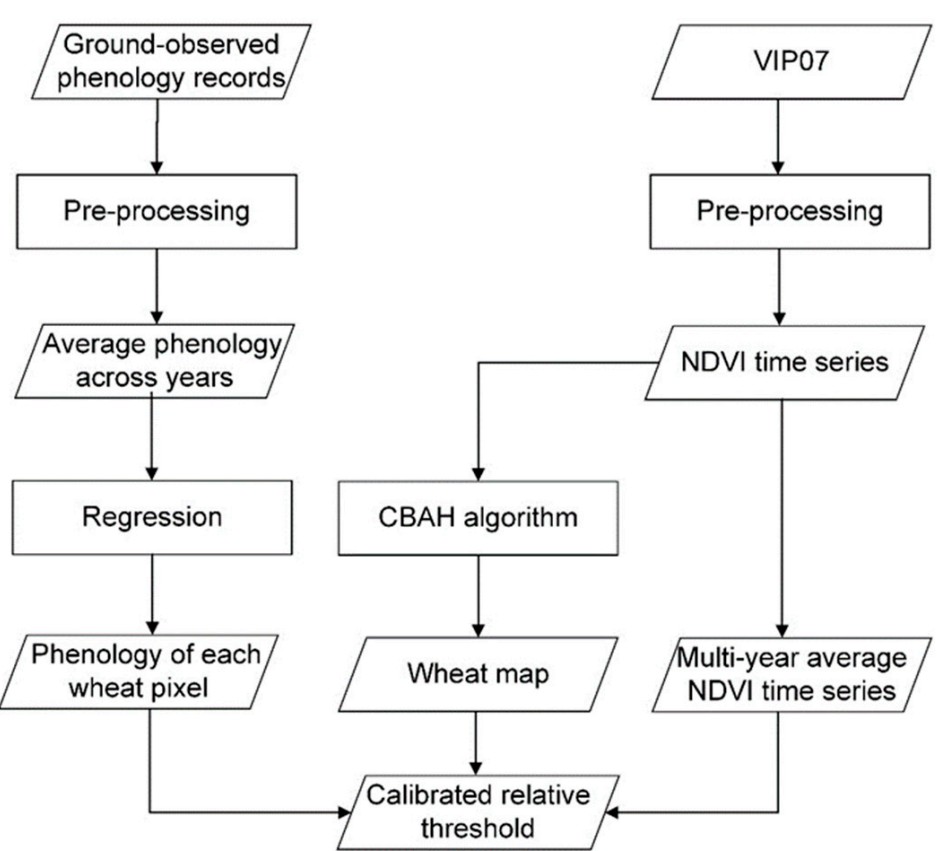

**Figure 2.** Flowchart of the calibrated relative threshold method (CRTM).

**Table 1.** Regression coefficients between phenological date and elevation, longitude, and latitude.

| Phenological Date | Coefficients | | | | R-Squared | *p*-Value |
|---|---|---|---|---|---|---|
| | Intercept | Altitude | Latitude | Longitude | | |
| Emergence | 404.092 | −0.012 | −4.233 | 0.364 | 0.734 | $p < 0.001$ |
| Tillering | 473.800 | −0.011 | −6.079 | 0.535 | 0.700 | $p < 0.001$ |
| Green-up onset | −114.400 | 0.012 | 2.215 | 0.767 | 0.546 | $p < 0.001$ |
| Jointing | −156.800 | 0.019 | 4.833 | 0.640 | 0.862 | $p < 0.001$ |
| Heading | −137.200 | 0.020 | 3.301 | 1.148 | 0.911 | $p < 0.001$ |
| Milky stage | −69.454 | 0.018 | 2.338 | 1.094 | 0.794 | $p < 0.001$ |
| Maturity | −57.935 | 0.019 | 2.450 | 1.083 | 0.855 | $p < 0.001$ |

As illustrated in Figure 3a, gray curves are annual NDVI time series for a specific wheat pixel. They were smoothed by the Savitzky–Golay (SG) filter [33] and were then averaged to generate a more stable multiyear average NDVI curve (green curve in Figure 3a). To remove outliers, only values between upper and lower quartiles were used for the multiyear average calculation [35,40]. The average NDVI curve was then accumulated to obtain the cumulative NDVI curve (black curve in Figure 3b) because previous studies have indicated that the cumulative NDVI curve performs better than the original NDVI curve in phenology extraction [41–43].

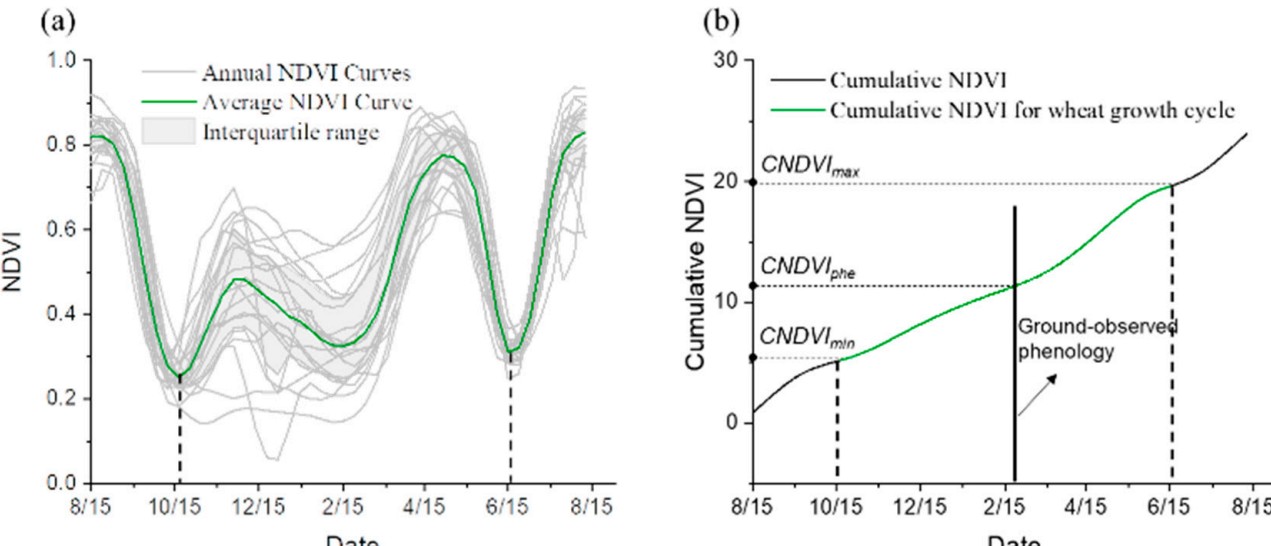

**Figure 3.** Schema to determine the relative threshold based on ground phenological observations. (**a**) represents annual NDVI curves for a specific wheat pixel, (**b**) illustrates how the relative threshold is determined based on ground phenological observations and cumulative NDVI curve.

The threshold for each phenological date was then determined using the cumulative NDVI curve based on the regression-estimated multiyear average phenological date (Figure 3b). The black thickened vertical line in Figure 3b represents a specific phenological date (e.g., green-up onset) of the wheat pixel. Table 1 shows the regression coefficients between different phenological dates of winter wheat and the elevation, longitude, and latitude.

According to the coefficients listed in Table 1, the multiyear average phenological date for each pixel of winter wheat was calculated from the altitude, latitude, and longitude (Equation (1)). In Equation (1), $Phe_{pixel}$ is the multiyear average phenological date of a wheat pixel. *Altitude*, *Latitude*, and *Longitude* are the altitude, latitude, and longitude at which the wheat pixel is located, respectively. $a_1$, $a_2$, and $a_3$ are regression coefficients, and $a_0$ is the intercept of the regression (Table 1):

$$Phe_{pixel} = a_0 + a_1 \cdot Altitude + a_2 \cdot Latitude + a_3 \cdot Longitude \tag{1}$$

The cumulative NDVI value corresponding to $Phe_{pixel}$ is expressed by $CNDVI_{phe_{pixel}}$; and the relative threshold for the phenological date of the specific wheat pixel can be obtained by Equation (2):

$$RThr_{cal} = \frac{CNDVI_{phe_{pixel}} - CNDVI_{min}}{CNDVI_{max} - CNDVI_{min}} \tag{2}$$

where $RThr_{cal}$ is the relative threshold for the target phenological date calibrated based on ground phenological observations, and $CNDVI_{min}$ and $CNDVI_{max}$ are cumulative NDVI values of the respective start and end dates of the wheat growth cycle. The start and end

dates of the wheat growth cycle can be obtained from the original NDVI curve and are set as the minimum value of NDVI before the winter season (vertical dashed line on the left in Figure 3a,b) and the minimum value of NDVI after the heading date (vertical dashed line on the right in Figure 3a,b).

According to the abovementioned process, the relative thresholds for different phenological dates of different wheat pixels can be calculated. The corresponding phenological dates can then be estimated from the annual cumulative NDVI time series of wheat pixels according to the relative threshold method (Equation (3)):

$$Phe_{date} = firstdaywhen(CNDVI \geq (CNDVI_{max} - CNDVI_{min}) \cdot RThr_{cal}) \tag{3}$$

After estimation of the seven key phenological dates, the lengths of the three growth periods (i.e., VGP, RGP, and VRGP) can be calculated according to Equations (4)–(6):

$$LVGP = Heading_{date} - Greenup_{date} \tag{4}$$

$$LRGP = Maturity_{date} - Heading_{date} \tag{5}$$

$$LVRGP = Maturity_{date} - Greenup_{date} \tag{6}$$

where LVGP is the length of the vegetative growth period, LRGP is the length of the reproductive growth period, and LVRGP is the total length of the vegetative and reproductive growth period. $Greenup_{date}$, $Heading_{date}$, and $Maturity_{date}$ represent the green-up onset date, heading date, and maturity date of winter wheat, respectively.

### 2.3.2. Calculation of Temporal Trends

Based on the estimated winter wheat phenological dates and growth period lengths for each year, the temporal trend of each phenological date (or growth period length) was determined as the coefficient using linear regression between the date and year.

### 2.3.3. Relationship between Phenology and Temperature

Correlation analysis was implemented to explore the relationship between the phenological date (or growth period) and its preseason temperature (or intraseasonal temperature) at each agrometeorological station. The preseason temperature for a specific phenological date is defined as the average temperature during the period from the current phenological date to its previous phenological date. For the first studied phenological date (i.e., emergence), the preseason temperature is defined as the average temperature between it and the previous month. The intraseasonal temperature for a specific growth period is defined as the average temperature during this growth period [7,44,45]. The phenological date (or growth period length) at each meteorological station was calculated as the average phenological date (or growth period length) of the wheat pixels contained in the Thiessen polygon of the corresponding meteorological station. The phenological date of wheat pixels used above was extracted by the CRTM method based on remotely sensed data [35,40]. The regression coefficient (*r*) is calculated as

$$r(X, Y) = \frac{Cov(X, Y)}{\sqrt{Var[X] \cdot Var[Y]}} \tag{7}$$

where *X* and *Y* correspond to the phenological date and preseason temperature, or the growth period length and intraseasonal temperature; $Cov(X, Y)$ is the covariance of *X* and *Y*; and $Var[X]$ and $Var[Y]$ represent the variance of *X* and *Y*, respectively.

## 3. Results

### 3.1. Performance of CRTM in Extracting Wheat Phenology

The phenological dates estimated by CRTM were statistically significantly correlated ($p < 0.001$) with ground-based phenological observations (Figure 4a,b,d–g). The correlation

coefficient was highest for the jointing date (R = 0.73), followed in descending order by the heading date (R = 0.72), maturity date (R = 0.70), milky stage date (R = 0.60), tillering date (R = 0.58), emergence date (R = 0.57), and green-up onset date (R = 0.23).

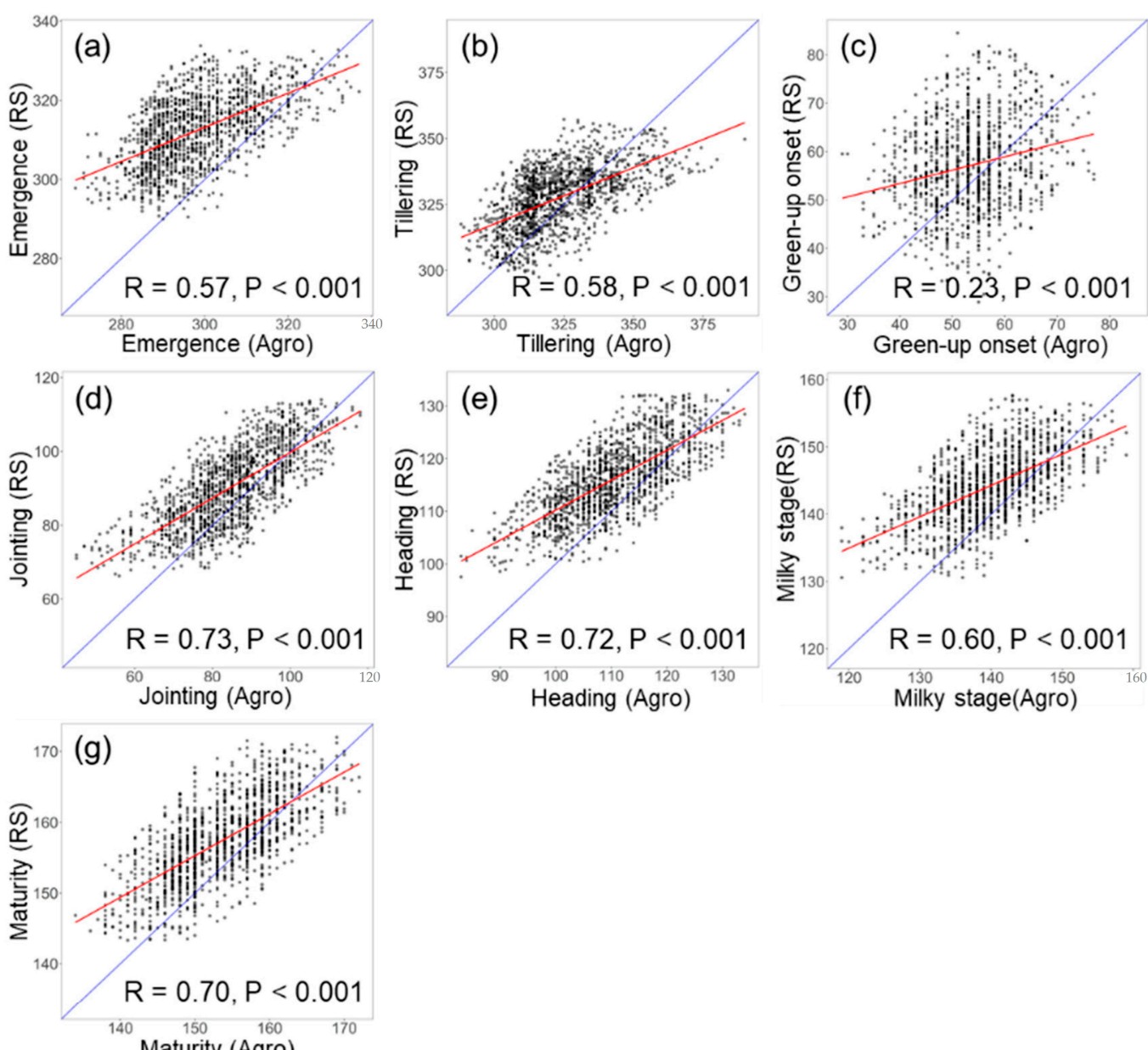

**Figure 4.** Comparison between CRTM-extracted phenological dates (Y-axis) and ground-based phenological observations (X-axis). (**a**) represents scatter plot for emergence, (**b**) represents scatter plot for tillering, (**c**) represents scatter plot for green-up onset, (**d**) represents scatter plot for jointing, (**e**) represents scatter plot for heading, (**f**) represents scatter plot for milky stage, and (**g**) represents scatter plot for maturity, respectively.

### 3.2. Spatial Patterns and Temporal Trends of Phenological Dates

Figure 5 shows the spatial patterns of the multiyear average phenological dates in the study area. Overall, these phenological dates showed a clear spatial pattern from the south to the north. Specifically, the emergence and tillering dates were earlier in the north and later in the south. The latest dates of these two phenological dates occurred in the southeastern and southwestern parts of the study area, respectively. By contrast, the other

five phenological dates (i.e., green-up onset, jointing, heading, milky stage, and maturity) were earlier in the southwestern part of the study area and gradually occurred later in the northeastern direction. The spatial variation of all of the wheat phenological dates was generally between 1 and 2 months in the study area.

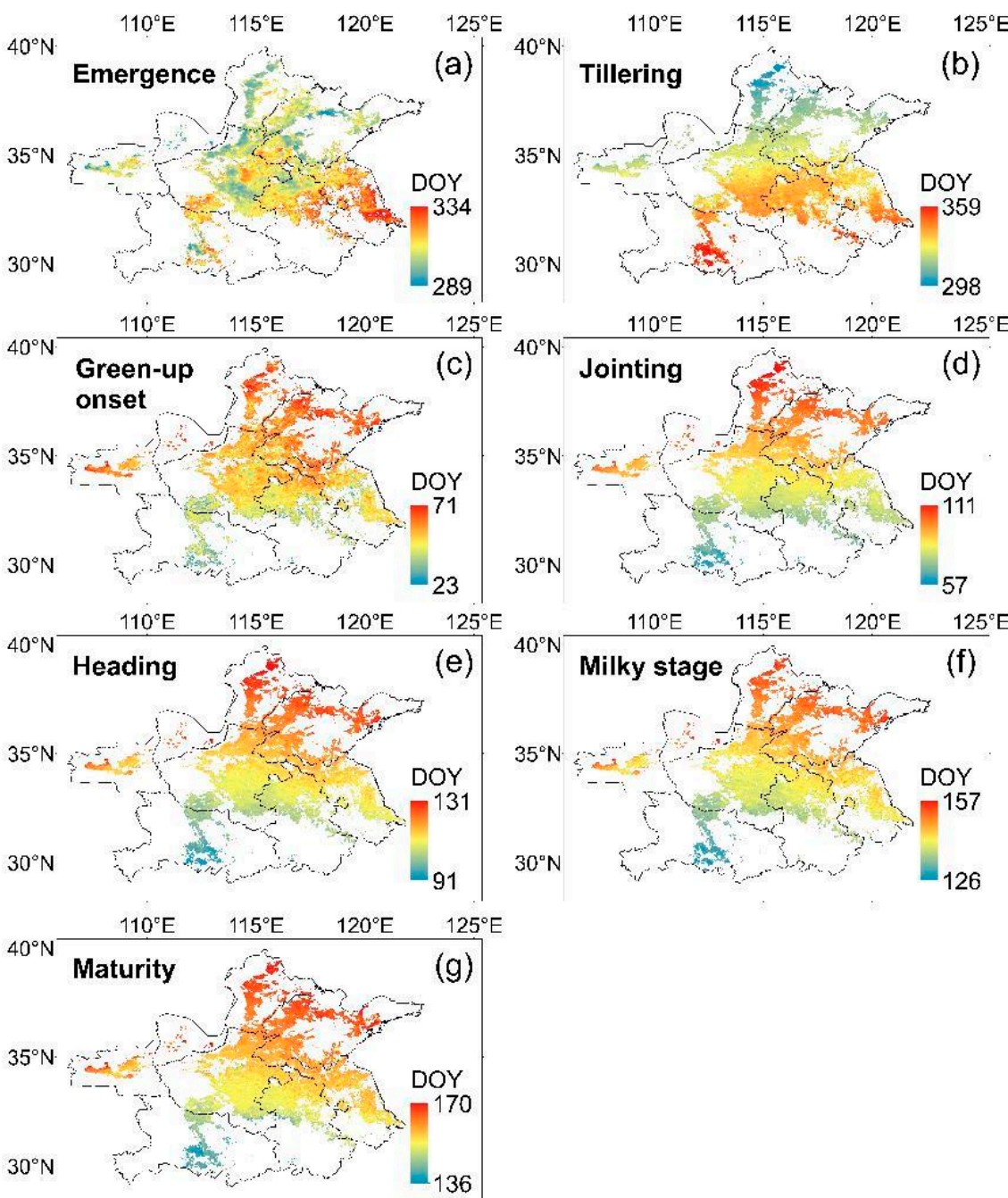

**Figure 5.** Spatial distribution of multiyear average phenology of winter wheat extracted from VIP data. DOY represents day of year. (**a**) illustrates spatial distribution of multiyear average phenology for emergence, (**b**) illustrates spatial distribution of multiyear average phenology for tillering, (**c**) illustrates spatial distribution of multiyear average phenology for green-up onset, (**d**) illustrates spatial distribution of multiyear average phenology for jointing, (**e**) illustrates spatial distribution of multiyear average phenology for heading, (**f**) illustrates spatial distribution of multiyear average phenology for milky stage, and (**g**) illustrates spatial distribution of multiyear average phenology for maturity, respectively.

Figure 6 shows the temporal trend and statistical significance of wheat phenological dates (e.g., Figure 6a,b). In general, the advanced or delayed trend of phenological dates had relatively similar spatial patterns, i.e., a significantly advanced trend in the south-central part of the study area and an insignificantly delayed trend in the eastern and northern parts (Figure 6). All phenological dates of winter wheat in the study area were dominantly advanced, with the greatest proportion of statistically significant advancement in the milky stage date and emergence date, accounting for 48.31% and 44.94%, respectively (Table 2). The significantly advanced proportion of the green-up onset and tillering date was the lowest, accounting for 38.83% and 40.17%, respectively. For the entire study area, the green-up onset, emergence, jointing, and heading dates had relatively large advances, with average values of 0.10, 0.09, 0.08, and 0.07 days per year, respectively ($p > 0.10$). The tillering and milky stage dates had relatively less advance, with average values of 0.05 and 0.03 days per year, respectively ($p > 0.10$). By contrast, there was a slight delay in the maturity date, with an average delay of 0.02 days per year ($p > 0.10$). In summary, the advance trend of phenological dates in the vegetative growth period was more pronounced than that in the reproductive growth period.

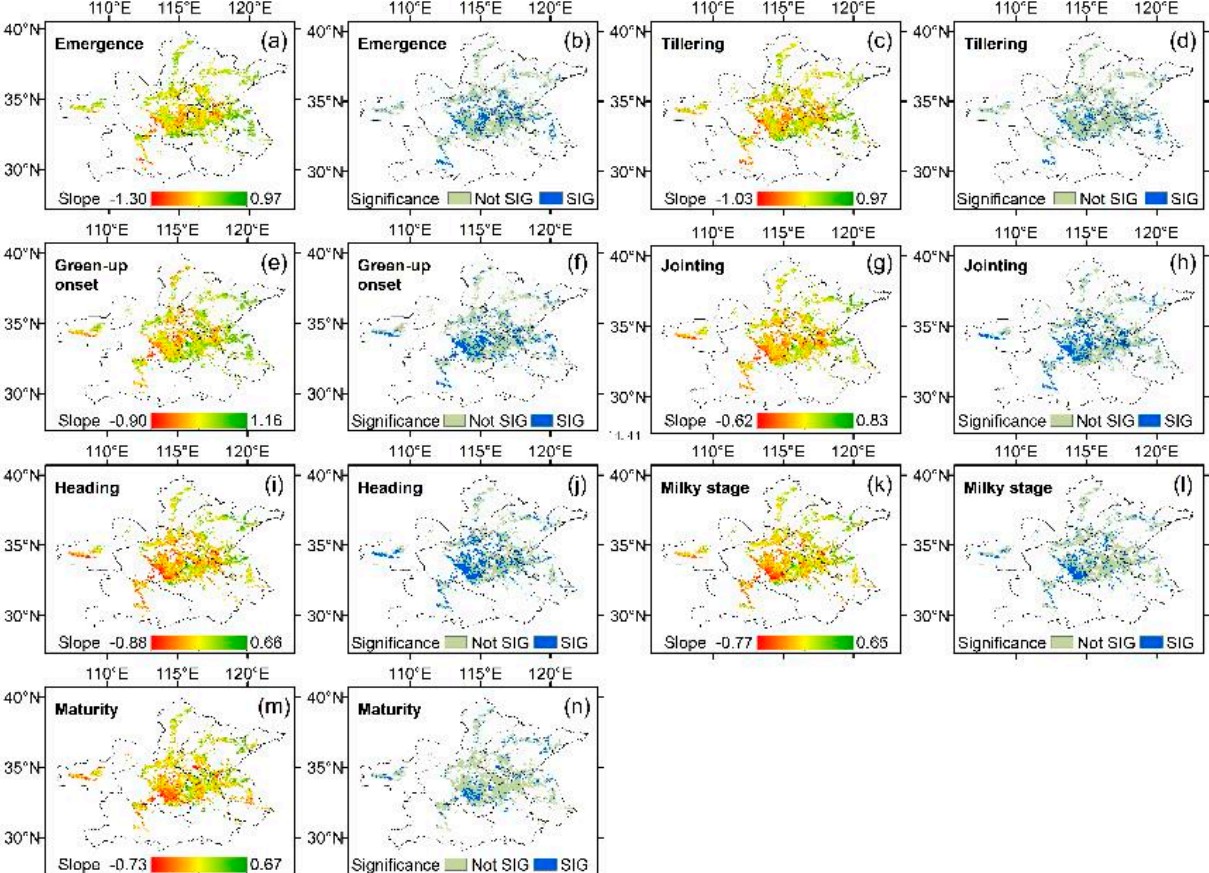

**Figure 6.** Phenological trends from 1981–1982 to 2015–2016. Negative slopes represent advanced phenological dates, and positive slopes, delayed dates. Wheat pixels with statistically significant trends ($p < 0.05$) are indicated in blue, and pixels without statistically significant trends are indicated in gray ($p > 0.05$). (**a,b**) represents trend and statistical significance of emergence, (**c,d**) represents trend and statistical significance of tillering, (**e,f**) represents trend and statistical significance of green-up onset, (**g,h**) represents trend and statistical significance of jointing, (**i,j**) represents trend and statistical significance of heading, (**k,l**) represents trend and statistical significance of milky stage, and (**m,n**) represents trend and statistical significance of maturity, respectively.

**Table 2.** Proportion of advance and delay of different phenological dates.

| PD | Advanced (%) | | Total (%) | Delayed (%) | | Total (%) | Average Trend (Day/Year) |
|---|---|---|---|---|---|---|---|
| | **S** | **NS** | | **S** | **NS** | | |
| **EMD** | 44.94 | 21.79 | 66.73 | 27.31 | 5.96 | 33.27 | −0.09 |
| **TID** | 40.17 | 12.29 | 52.45 | 37.86 | 9.68 | 47.55 | −0.05 |
| **GUD** | 38.83 | 24.16 | 62.99 | 28.59 | 8.42 | 37.01 | −0.10 |
| **JTD** | 40.62 | 30.21 | 70.83 | 23.59 | 5.58 | 29.17 | −0.08 |
| **HD** | 42.52 | 31.12 | 73.64 | 21.58 | 4.78 | 26.36 | −0.07 |
| **MKD** | 48.31 | 19.57 | 67.88 | 26.93 | 5.20 | 32.12 | −0.03 |
| **MTD** | 43.56 | 10.70 | 54.26 | 37.83 | 7.91 | 45.74 | 0.02 |

**Note.** PD represents phenological dates; EMD, TID, GUD, JTD, HD, MKD, and MTD represent dates of emergence, tillering, green-up onset, jointing, heading, milky stage, and maturity, respectively. S represents statistically significant, and NS represents not statistically significant.

### 3.3. Spatial Patterns and Temporal Trends of Wheat Growth Periods

Figure 7 shows the spatial patterns of multiyear averages for different growth periods (i.e., VGP, RGP, and VRGP) in the study area. It can be seen that the reproductive growth period (RGP) was most clearly characterized by a gradual change from south to north (Figure 7b), followed by the vegetative growth period (VGP, Figure 7a), while the total length of these two growth periods (vegetative and reproductive growth period, VRGP) did not show a clear spatial pattern (Figure 7c). Specifically, the VGP was relatively longer in the north and shorter in the south, while the RGP was relatively longer in the south and shorter in the north. Since VGP and RGP had opposite trends in the north–south direction, the total length of the two growth periods (VRGP) did not show a clear north–south trend (Figure 7c). Differences in the length of the growth periods reached 1 month throughout the study area.

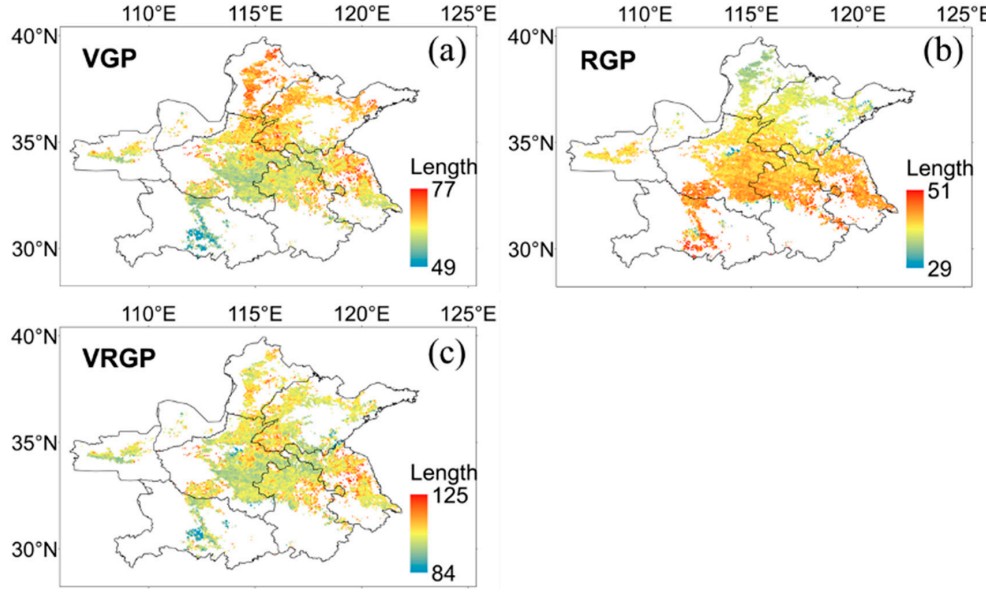

**Figure 7.** Spatial distribution of multiyear averages of different growth periods extracted from VIP data. VGP represents the vegetative growth period, RGP represents the reproductive growth period, and VRGP represents the vegetative and reproductive growth periods combined. (**a**) represents spatial patterns of multiyear averages for VGP, (**b**) represents spatial patterns of multiyear averages for RGP, and (**c**) represents spatial patterns of multiyear averages for VRGP, respectively.

Figure 8 shows the temporal trend and statistical significance of the length of different growth periods (i.e., LVGP, LRGP, and LVRGP). Overall, most wheat pixels in the study area were dominated by an extending trend in the three growth periods, mostly the

LRGP (length of the reproductive growth period), followed by the LVRGP (length of the vegetative and reproductive growth periods combined) and LVGP (length of the vegetative growth period). The reproductive growth period had the largest statistically significant extending proportion, accounting for about 43.63%, while the vegetative growth period had the least statistically significant extending proportion, accounting for 39.11% (Table 3). Although the proportion of the extended vegetative growth period was greater than that of shortening, the difference was not significant, indicating that nearly half of the winter wheat in the study area still showed a shortened trend of the vegetative growth period. Pixels with a shortened vegetative growth period were mainly located in the southeastern part of the study area (Figure 8a). Averaged over the entire study area, the LVGP extended by an average of 0.03 days per year ($p > 0.10$), the LRGP by an average of 0.09 days per year ($p < 0.001$), and the LVRGP by an average of 0.12 days per year ($p > 0.01$).

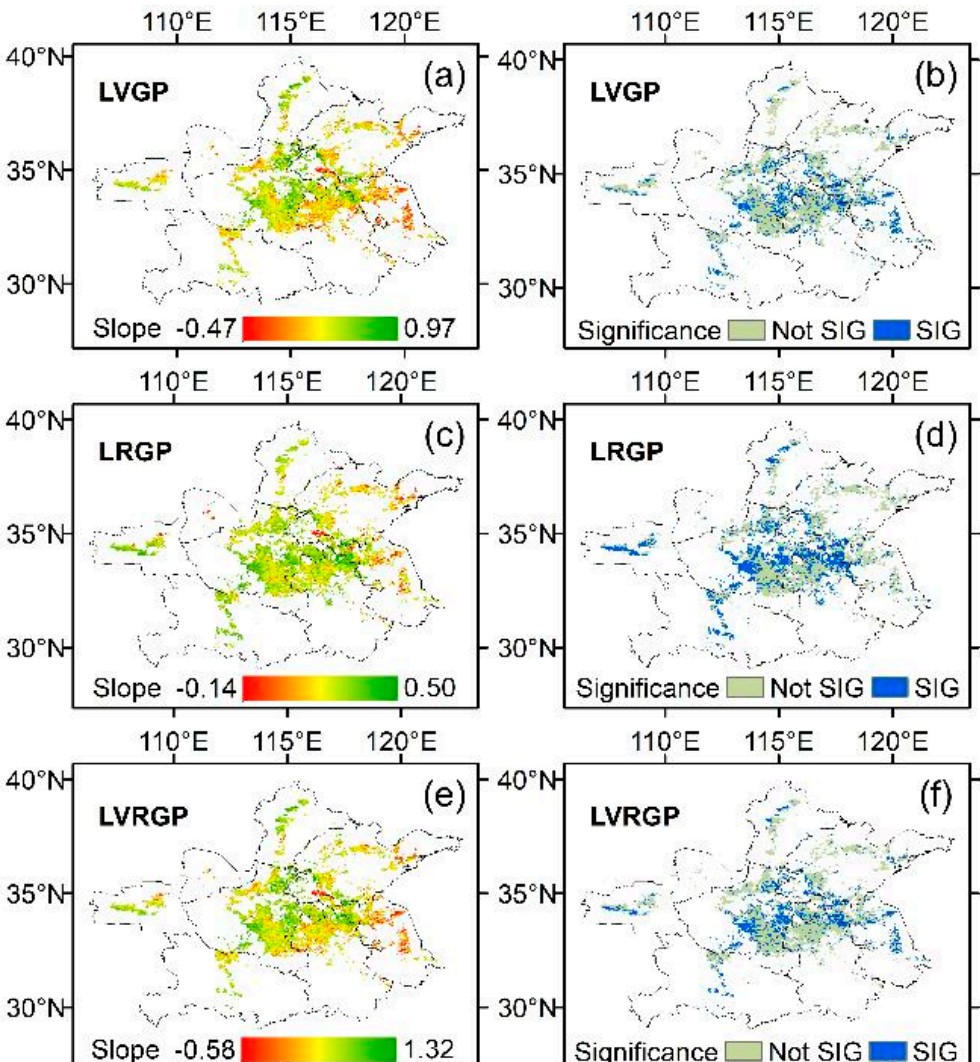

**Figure 8.** Trends of the growth period length (first column) calculated from 1981–1982 to 2015–2016 and its statistical significance (second column). The red color represents a shortened or decreasing trend, while the green color represents an extended or increasing trend. Negative slopes represent shortened trends, while positive slopes represent extended trends. Wheat pixels with statistically significant trends ($p < 0.05$) are indicated in blue, and pixels without statistical significance are indicated in gray ($p > 0.05$). (**a**,**b**) represents trend and statistical significance for LVGP, (**c**,**d**) represents trend and statistical significance for LRGP, and (**e**,**f**) represents trend and statistical significance for LVRGP, respectively.

**Table 3.** Proportion of shortened and extended lengths for different growth periods.

| Growth Periods | Shortened (%) | | Total (%) | Extended (%) | | Total (%) | Average Trend (Day/Year) |
|---|---|---|---|---|---|---|---|
| | S | NS | | S | NS | | |
| VGP | 32.50 | 11.06 | 43.56 | 39.11 | 17.33 | 56.44 | 0.03 |
| RGP | 18.24 | 2.20 | 20.44 | 43.63 | 35.93 | 79.56 | 0.09 |
| VGRP | 27.73 | 7.57 | 35.30 | 40.92 | 23.77 | 64.70 | 0.12 |

**Note.** S represents statistically significant, and NS represents not statistically significant.

### 3.4. Relationships between Phenological Metrics and Temperature

3.4.1. Relationships between Phenological Dates and Preseason Temperatures

The relationships between the phenological dates and the corresponding preseason temperature are shown in Figure 9. Overall, the phenological dates and corresponding preseason temperatures were mainly negatively correlated, except for the green-up onset date. The highest proportion of sites with statistically significant negative correlations ($p < 0.05$) with temperature was at the milky stage (36.05%), followed by maturity (35.89%), heading (33.01%), tillering (17.69%), jointing (16.34%), emergence (7.48%), and green-up onset (3.23%). The sites with statistically significant negative correlations ($p < 0.05$) at the milky stage were mainly located in the central and northern parts of the study area (Figure 9f), while those with negative correlations at maturity and heading were mainly located in the south-central part of the study area (Figure 9g,e). For the other phenological dates (i.e., emergence, tillering, green-up onset, and jointing), sites with statistically significant negative correlations ($p < 0.05$) were sparsely distributed in different parts of the study area (Figure 9a–d).

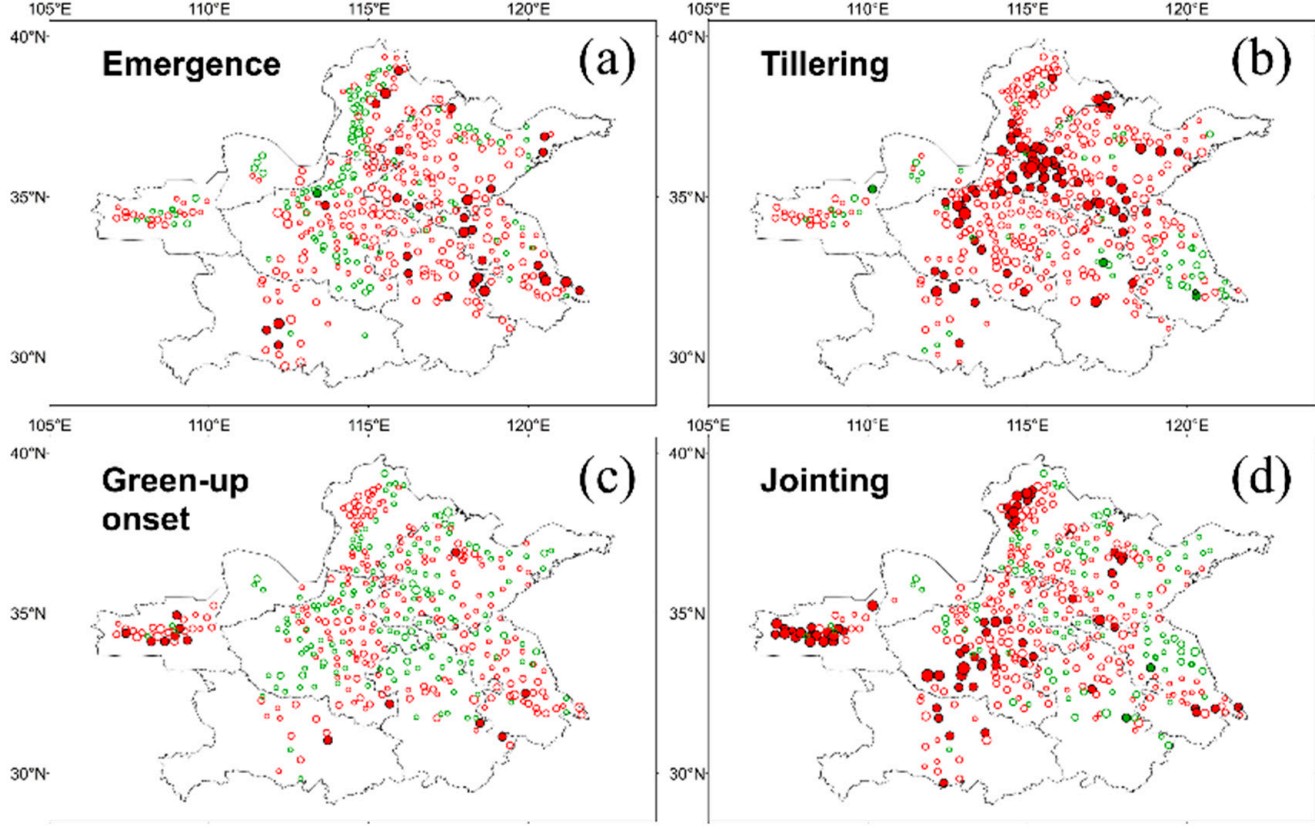

**Figure 9.** *Cont.*

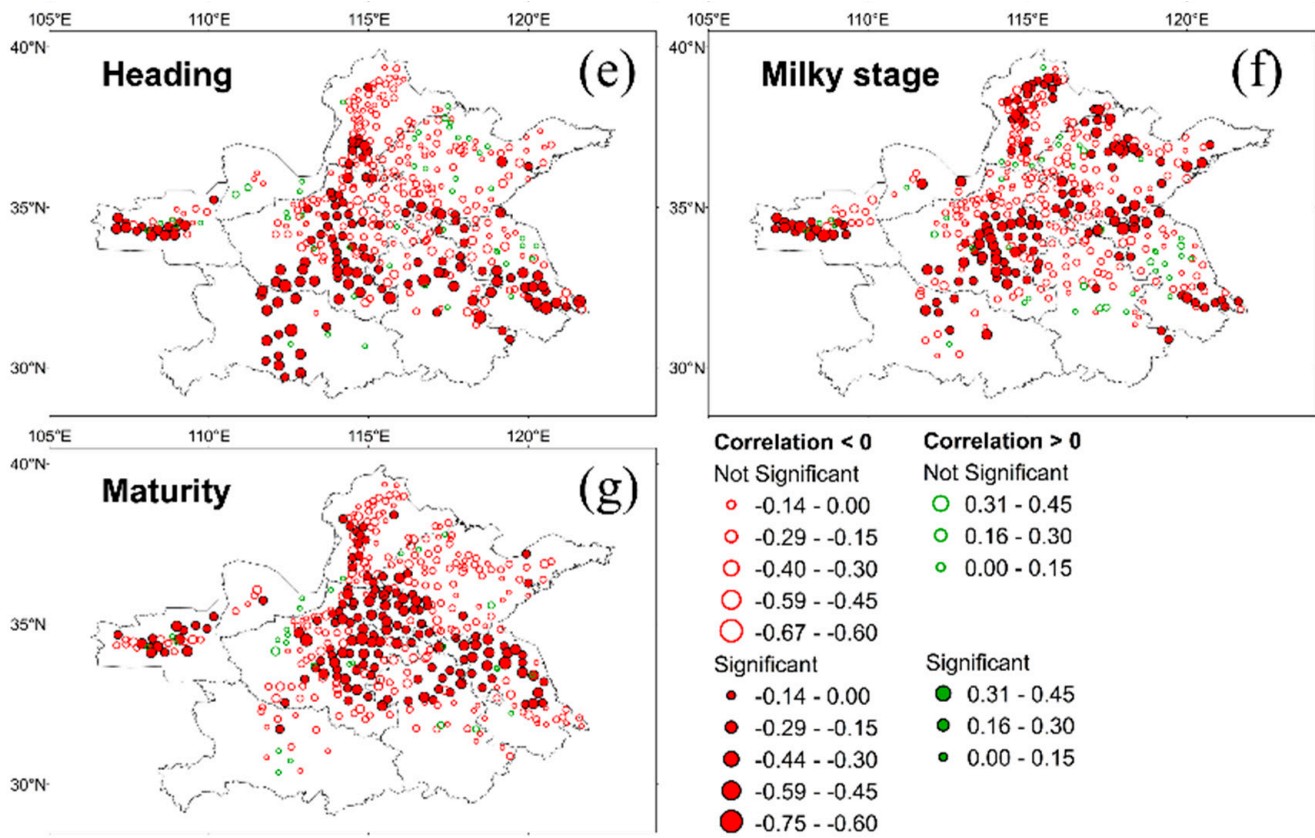

**Figure 9.** Correlation between phenological dates and corresponding preseason temperature from 1981 to 2016. Red circles represent negative correlations, and green circles represent positive correlations. Hollow circles indicate statistically insignificant correlations ($p > 0.05$), while solid circles indicate statistically significant correlations ($p < 0.05$). (**a**) represents correlation for emergence, (**b**) represents correlation for tillering, (**c**) represents correlation for green-up onset, (**d**) represents correlation for jointing, (**e**) represents correlation for heading, (**f**) represents correlation for milky stage, and (**g**) represents correlation for maturity, respectively.

### 3.4.2. Relationships between the Growth Period Length and Intraseasonal Temperatures

The correlations between different growth period lengths and intraseasonal temperatures are shown in Figure 10. In general, the length of the vegetative growth period (VGP) was mainly negatively correlated with intraseasonal temperatures, while the length of the reproductive growth period (RGP) and the length of the vegetative and reproductive growth period (VRGP) were mainly positively correlated. The highest proportion of sites showed a statistically significant negative correlation ($p < 0.05$) between the intraseasonal temperature and the vegetative growth period (VGP, 21.17%), followed by the vegetative and reproductive growth period (VRGP, 10.40%). They were mainly distributed in the southeastern part of the study area (Figure 10a,c). For the reproductive growth period, the proportion of sites with a statistically significant positive correlation ($p < 0.05$) with intraseasonal temperatures was 17.94%, and they were mainly distributed in the northern and central parts of the study area (Figure 10b).

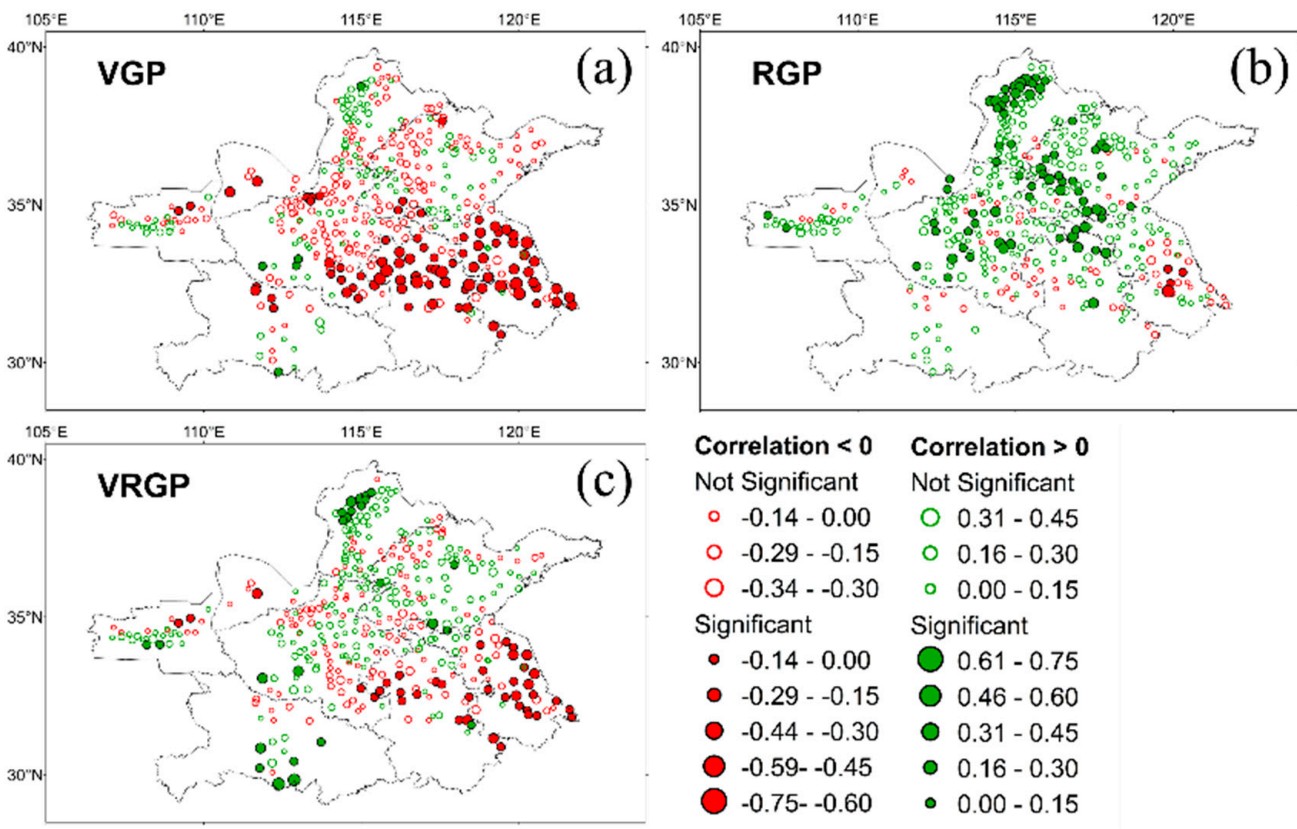

**Figure 10.** Correlations between different growth periods and intraseasonal temperatures from 1981 to 2016. Red circles represent negative correlations, and green circles represent positive correlations. Solid and hollow circles indicate statistically significant ($p < 0.05$) and insignificant ($p > 0.05$) correlations, respectively. (**a**) represents correlation for VGP, (**b**) represents correlation for RGP, and (**c**) represents correlation for VRGP, respectively.

## 4. Discussion

### 4.1. Advantages of CRTM and Its Limitations

When using the relative threshold method to extract phenological dates, the choice of thresholds in existing studies is usually empirical, and the same empirical threshold is used throughout the study area. This method of determining the threshold is relatively simple, but an empirical threshold (e.g., 10%, 20%, or 50%) may not be optimal [25,28,29] because the threshold is spatially variable. Figure 11 shows the spatial distribution of the calibrated relative threshold (CRT) for each phenological date determined based on ground-based phenological observations. The relative thresholds of all phenological dates had large spatial variations across the study area.

Figure 12 shows the histogram of relative thresholds (CRT) for different phenological dates. As the histogram of the relative thresholds at emergence was narrow (red curve in Figure 12) and fluctuated between 0.02 and 0.05 (Figure 11a), a fixed relative threshold (i.e., taking the average of 0.02 and 0.05) was used throughout the study area when extracting the emergence date using the relative threshold method. For the other phenological dates (i.e., tillering, green-up onset, jointing, heading, milky stage, and maturity), our result showed there was no globally fixed threshold for extracting the phenological date across the study area due to the wide spatial variation of the relative thresholds (Figure 11). Therefore, using a fixed empirically based relative threshold (e.g., 10%, 20%, or 50%) to extract the phenological date would introduce large bias and uncertainty, especially over large areas.

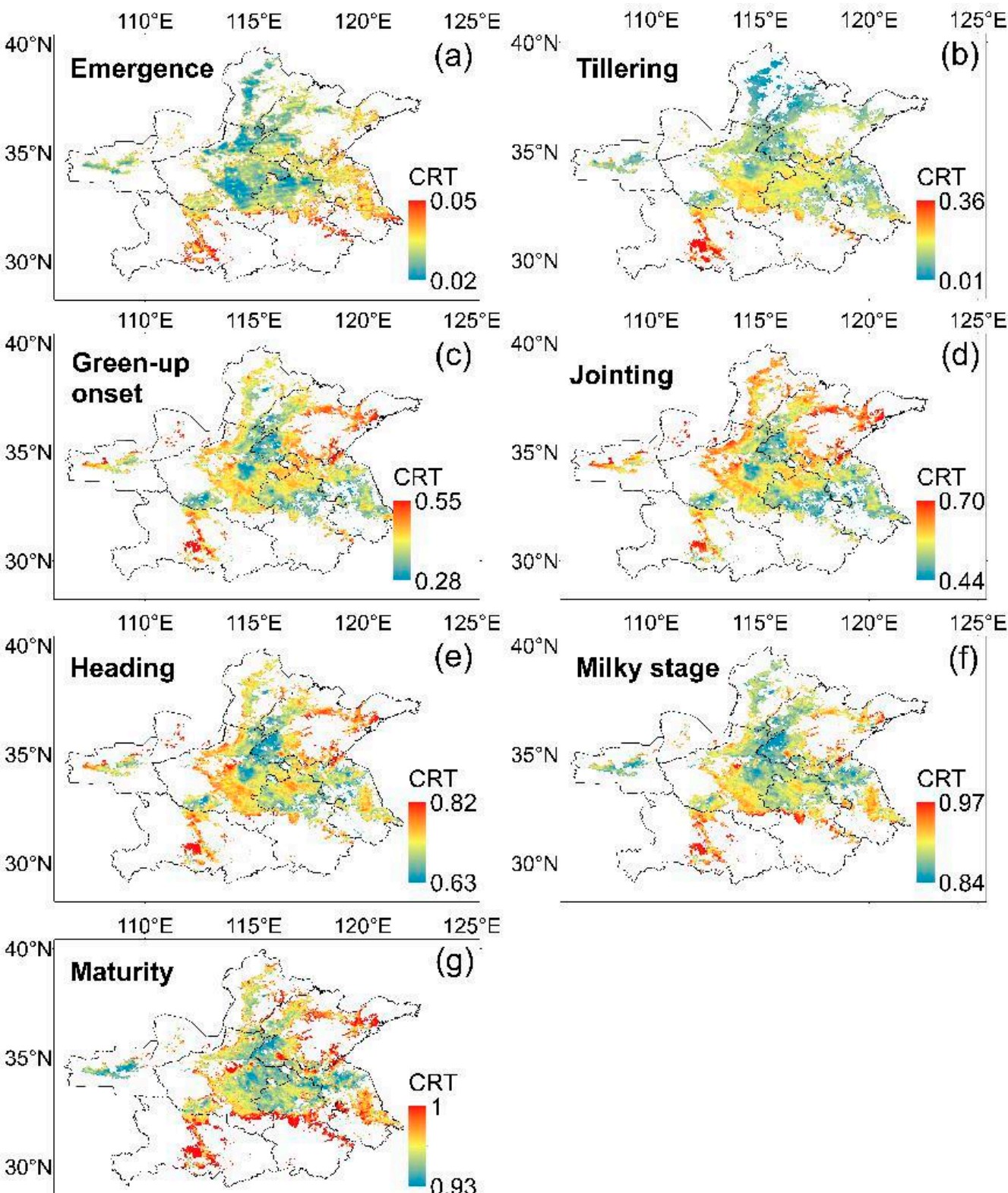

**Figure 11.** Spatial distribution of the calibrated relative threshold (CRT) for different phenological dates based on ground-based phenological observations. (**a**) illustrates spatial distribution of CRT for emergence, (**b**) illustrates spatial distribution of CRT for tillering, (**c**) illustrates spatial distribution of CRT for green-up onset, (**d**) illustrates spatial distribution of CRT for jointing, (**e**) illustrates spatial distribution of CRT for heading, (**f**) illustrates spatial distribution of CRT for milky stage, and (**g**) illustrates spatial distribution of CRT for maturity, respectively.

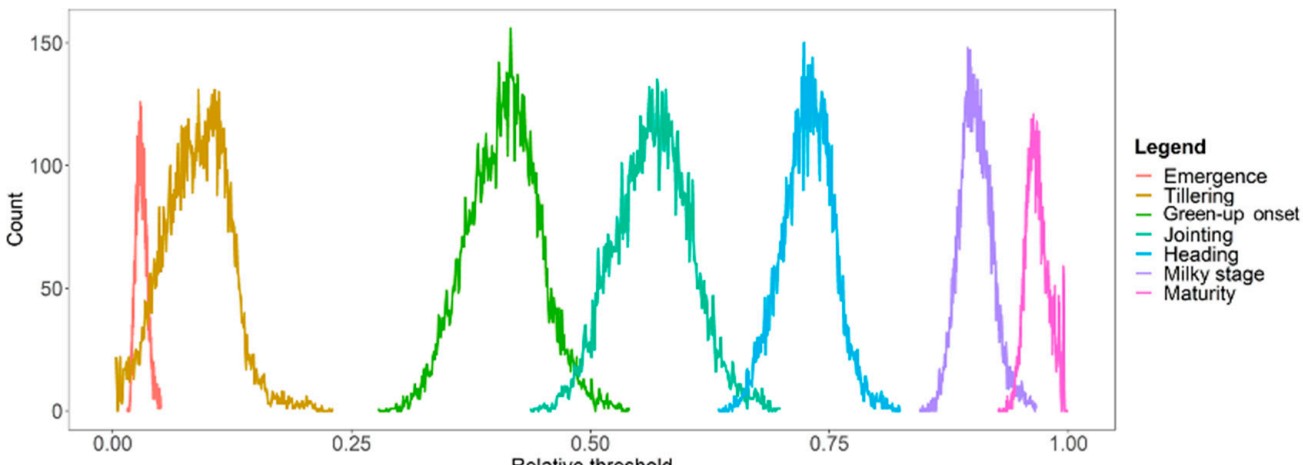

**Figure 12.** Histogram distribution of the relative thresholds of wheat phenology.

By contrast, the newly proposed CRTM in this study can minimize the bias and uncertainty caused by unreasonable threshold settings when extracting phenological dates. Comparisons with ground-based phenological observations (Figure 4) suggest that the newly proposed CRTM can effectively extract the key phenological dates of winter wheat from the remotely sensed NDVI time series. Although there were absolute quantitative differences between the phenological dates extracted by CRTM and ground phenological observations, the consistency of the relative change trends can be guaranteed, which indicates that phenological dates estimated by CRTM are appropriate for further exploring the long-term spatial–temporal patterns.

Furthermore, it has been shown that cumulative NDVI curves can better overcome the interference of natural variability in phenology extraction [41–43]. Therefore, the original NDVI curves were replaced with cumulative NDVI curves when using the calibrated relative threshold method. This also had the significant advantage that when determining the relative thresholds for different phenological dates from the original NDVI curves, it was necessary to first identify the monotonically increasing or decreasing segments of NDVI curves where different phenological dates occurred. Such segments are the rising segment of NDVI before winter, the falling segment of NDVI during overwintering, the rising segment of NDVI in spring, and the falling segment of NDVI in post-heading stages. As NDVI curves of winter wheat have complex bimodal characteristics and are susceptible to various noises (e.g., cloud contamination), it significantly increases the complexity and computational effort of searching feature points in the original NDVI curves. In addition, the different ascending and descending segments where different phenological dates are located further increase the complexity and computational effort of extracting the phenological dates from the original NDVI curves. By contrast, the cumulative NDVI curve is a monotonically increasing curve with very simple morphological characteristics, so the relative threshold method can be easily implemented based on the same rising segment of the cumulative NDVI curve, and only the start and end dates of the winter wheat growth period need to be determined from the original NDVI curve, which greatly reduces the complexity and computational effort of extracting different phenological dates. Furthermore, the monotonically increasing characteristics of the cumulative NDVI curve are not affected by the morphology of the original NDVI curve and can therefore be directly applied to the phenology extraction for different crop types and different phenological dates.

The newly proposed CRTM also has some limitations because it first requires ground-based phenological observations when determining the relative thresholds for different phenological dates in different locations. Although this can improve the accuracy of phenology extraction, it is a challenge for areas lacking phenological observations. Moreover, if winter wheat encounters catastrophic events (e.g., spring frost, dry-hot wind, pest in-

festation) that cause a significant decrease in NDVI, it will impose an adverse impact on the calculation of the cumulative NDVI curves, thus bringing some uncertainty to the phenology extraction. For example, the occurrence of frost in early spring poses a threat to winter wheat and brings large fluctuations in NDVI curves (Figure 3a) [35], which may be responsible for the relatively low coefficients between the CRTM-estimated and ground-observed green-up onset date (Figure 4b). Therefore, the effects of catastrophic events on winter wheat phenology and how such effects are reflected in the changes in the original and cumulative NDVI curves should be further investigated in future studies.

*4.2. Comparisons with Existing Studies*

Liu and Wang [20] investigated the trends of the green-up onset date, heading date, and the length between the green-up onset date and heading date (defined as STAGE) in North China from 1982 to 2015 based on GIMMS data. Their results showed that the average advancement of green-up onset and of the heading date in the study area was 1.8 days/decade and 1.3 days/decade, respectively. The green-up date advanced faster than the heading date, resulting in a lengthened STAGE by 1 day/decade. A similar pattern was found in our study, i.e., the green-up onset date was advanced by an average of 1.0 days/decade, and the heading date was advanced by an average of 0.7 days/decade. The green-up onset date advanced faster than the heading date, resulting in an average extension of the vegetative growth period (i.e., STAGE in Liu and Wang's study) by 0.3 days/decade. In addition, our study found that the advanced trend of the green-up onset date was the most pronounced (1.0 days/decade) among seven studied phenological dates; the proportion of wheat pixels with significantly advanced green-up onset date was only 38.83%, which was lower than the proportion of wheat pixels with significantly advanced jointing and heading stages (i.e., 40.62% and 42.52%, respectively). Wang et al [13] showed that the reproductive growth period of winter wheat on the North China Plain was prolonged during the period 1981–2012. A prolonged trend was observed in 87% of the sites with an average of 0.6 days/decade. Our study found that nearly 43.63% of winter wheat in the study area exhibited a significantly extended reproductive growth period, averaging 0.9 days/decade across the study area, which is longer than that in the study conducted by Wang et al. [13] based on site observations.

Existing studies on winter wheat phenology have always differed in the selection of phenological dates, the period studied, the data sources used, and the phenology extraction methods [7,13,15], making the results vary from study to study and making it difficult to completely describe the spatial and temporal variation of each phenological date. Compared with existing studies that targeted a single or few phenological dates (or growth periods), this study systematically investigated the spatial and temporal variation of seven phenological dates and three growth periods using a unified data source and phenology extraction method, which is more conducive to determining the spatiotemporal characteristics of winter wheat phenology as a whole. For example, this study found that the pre-wintering phenological dates of winter wheat (i.e., emergence and tillering) showed the spatial distribution of late in the south and early in the north, while the post-wintering phenological dates (i.e., green-up onset, jointing, heading, and maturity) showed the spatial distribution of early in the south and late in the north. Similarly, the vegetative growth period of winter wheat showed the spatial distribution of short in the south and long in the north, while the reproductive growth period showed the spatial distribution of long in the south and short in the north.

In terms of temporal trends, our study found that the advanced or delayed trend of each phenological date had similar spatial patterns. For example, wheat phenological dates in the central part of the study area mostly showed a more significantly advanced trend, while phenological dates in the eastern and northern parts of the study area mostly showed a non-significant delayed trend. This co-movement effect (simultaneous advance or delay) between different phenological dates is likely to be related to the interaction between different phenological dates. For example, Wu et al. (2019) [46] suggested that

winter wheat exhibited a positive correlation between green-up and the heading date as well as between heading and the maturity date, where 1 day earlier green-up date may lead to a 0.57-day earlier heading date, and 1 day earlier heading date may lead to a 0.60-day earlier maturity date. These patterns were difficult to find in previous studies that focused only on a single or a few phenological dates.

By contrast, with respect to the relationship between wheat phenology and temperature, existing studies have shown that increasing temperature accelerates the growth process of winter wheat, resulting in an advanced trend of most phenological dates, as well as a shortened vegetative growth period and prolonged reproductive growth period [7,12,13]. Our study found that the tillering, jointing, heading, milky stage, and maturity of winter wheat were negatively correlated with the preseason temperature, and the length of the reproductive growth period was positively correlated with the intraseasonal mean temperature, which is consistent with existing studies. Although emergence, green-up onset, and jointing were predominantly negatively correlated with pre-season mean temperature, there were still a number of sites that showed a non-significant positive correlation, suggesting that the conclusion that warmer temperatures accelerate the wheat growth process may not be applicable for all phenological dates. Our study also found that the relationship between the vegetative growth period and the corresponding intraseasonal temperature was not spatially stable, i.e., the increase in the intraseasonal temperature in the southern part of the study area shortened the length of the vegetative growth period, while the increase in the intraseasonal temperature in the north-central part prolonged the vegetative growth period. From the relationship between the phenological dates and the corresponding preseason temperature, it can be seen that this was mainly caused by the different trends and extent of changes in the green-up onset and heading date of winter wheat in different regions under the background of climate warming.

## 5. Conclusions

In this study, we propose a calibrated relative threshold method that can fully use the ground-based phenology observations from previous years to extract winter wheat phenology. This method can determine the relative thresholds for different phenological dates in different locations based on the ground phenological observations and can thus minimize the bias and uncertainty caused by unreasonable threshold settings when extracting phenology by the relative threshold method. Based on this method, we found that the pre-wintering phenological dates of winter wheat (i.e., emergence and tillering) showed the spatial distribution of late in the south and early in the north, while the post-wintering phenological dates (i.e., green-up, jointing, heading, milky stage, and maturity) showed the spatial distribution of early in the south and late in the north. Winter wheat in the northern part of the study area has a longer vegetative growth period and a shorter reproductive growth period than winter wheat in the south. In terms of temporal trends, all seven studied phenological dates were predominantly advanced, with the most significant advancement in the green-up, emergence, jointing, and heading stages. The vegetative growth period of winter wheat was mainly shortened in the southeastern part of the study area and prolonged in the north-central part of the study area. The reproductive growth period of winter wheat was predominantly prolonged throughout the study area. As far as the relationship with temperature is concerned, the winter wheat phenological dates mainly advanced when the pre-season mean temperature increased. When the intraseasonal mean temperature increased, the vegetative growth period tended to shorten in the south-central part of the study area and extended in the northern part of the study area, while the reproductive growth period tended to extend throughout the study area. These results provide a reference for improving the management of winter wheat and crop production.

**Author Contributions:** J.C. conceived the idea and designed the research. S.W. conducted the experiments and drafted the manuscript. M.S., C.W. and J.C. revised the manuscript. T.S., L.L., L.Z. and Q.D. contributed to the scientific discussion of the article. All authors have read and agreed to the published version of the manuscript.

**Funding:** This research was funded by the National Natural Science Foundation of China (42101370), the Provincial Natural Science Foundation of Fujian Province (No. 2022J05244), the Fundamental Research Funds for the Central Universities, the Fujian Social Science Foundation Project (FJ2021C090), and the Fujian Educational Research Project for Young and Middle-aged Teachers (JAT200423).

**Data Availability Statement:** Not applicable.

**Conflicts of Interest:** The authors declare no conflict of interest.

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
