# Peer review of "Characterizing Spatiotemporal Patterns of Winter Wheat Phenology from 1981 to 2016 in North China by Improving Phenology Estimation"

_remotesensing, doi:10.3390/rs14194930_

Round 1
Reviewer 1 Report
The study provides an improved method to estimate phenological dates of winter wheat in China. The new method specified the relative threshold for each phenological date in different locations according to ground phenological observations, reducing the bias and uncertainties caused by the previous unrealistic threshold settings. Based on the improved phenological dates, the authors analyzed spatial and temporal variations in different phenological dates of the winter wheat in China. They found different warming effects on the growth period across different regions. Overall, the study significantly improved estimation of multiple phenological dates from satellite data and provided important references about changes and drivers of phenological dates of wheat in China during the recent warming. Here are my concerns about the paper before it can be published.
In the correlation analysis, I was wondering whether the authors ever use the remotely sensed results to investigate the impact of temperature on the phenological date. I just found the results derived from ground observations (Fig. 8).
In many places of the manuscript, only one word “significant” or “significance” was used, but I was wondering “statistically significant” or “statistical significance” would be more professional to describe the results.
Lines 159–162: I suggest the authors provide the purpose in using the topographic data here, even though how to use the altitude data is mentioned below.
Lines 279–284: I suggest the authors offer references for the definitions of the temperature so that readers could be clearer about why the authors use the temperature defined in the manuscript.
Line 353: “spatial trend” seems wired. I suggest using “spatial pattern”.
Author Response
Reviewer 1
The study provides an improved method to estimate phenological dates of winter wheat in China. The new method specified the relative threshold for each phenological date in different locations according to ground phenological observations, reducing the bias and uncertainties caused by the previous unrealistic threshold settings. Based on the improved phenological dates, the authors analyzed spatial and temporal variations in different phenological dates of the winter wheat in China. They found different warming effects on the growth period across different regions. Overall, the study significantly improved estimation of multiple phenological dates from satellite data and provided important references about changes and drivers of phenological dates of wheat in China during the recent warming. Here are my concerns about the paper before it can be published.
In the correlation analysis, I was wondering whether the authors ever use the remotely sensed results to investigate the impact of temperature on the phenological date. I just found the results derived from ground observations (Fig. 8).
Response: Thank you for your comments. We used phenological dates extracted from remotely sensed date in the correlation analysis. But it was not clearly explained in the original manuscript in lines 285-288. In the revised manuscript, we have added the information of using the remotely sensed results in the correlation analysis (Page 12, lines 299-300).
“Phenoloical date of wheat pixels use above was extracted by CRTM method based on remotely sensed data.”
In many places of the manuscript, only one word “significant” or “significance” was used, but I was wondering “statistically significant” or “statistical significance” would be more professional to describe the results.
Response: Thank you for your helpful suggestions. We have replaced “significant” and “significance” by “statistically significant” and “statistical significance” in the revised manuscript following your advice.
Lines 159–162: I suggest the authors provide the purpose in using the topographic data here, even though how to use the altitude data is mentioned below.
Response: Thank you for your suggestions. The purpose of using the topographic data has been added in lines 164-166 and 220-223 in the revised manuscript.
“Latitude, longitude, and altitude extracted from the topographic data was then used to estimate the phenological date of each wheat pixel (more details in Section 2.3.1).”
“Therefore, the phenological date of each wheat pixel was estimated by regressing the multiyear average phenological date of all station across the study area (i.e., 378 stations) against the station’s latitude, longitude, and elevation (Table 1) based on topographic data (i.e., GMTED2010).”
Lines 279–284: I suggest the authors offer references for the definitions of the temperature so that readers could be clearer about why the authors use the temperature defined in the manuscript.
Response: Thank you for your suggestions. We have offered three references for the definitions of the temperature in line 296 in the revised manuscript.
Line 353: “spatial trend” seems wired. I suggest using “spatial pattern”.
Response: Thank you for your helpful suggestion. The “spatial trend” in line 370 has been replaced with “spatial pattern” as you suggested.

Reviewer 2 Report
The manuscript aims to extract spatial and temporal variability of winter wheat phenology in Northern China. A calibrated threshold value of accumulated NDVI was proposed to extract different phenological events in winter wheat. Spatial and temporal variability of winter wheat phenology and theirs response to temperature were further assessed. In fact, extracting phenological metrics based on accumulated NDVI is not new, which can be found in the previous studies such as, Wu, Chaoyang, Xuehui Hou, Dailiang Peng, Alemu Gonsamo, and Shiguang Xu. "Land surface phenology of China's temperate ecosystems over 1999–2013: Spatial–temporal patterns, interaction effects, covariation with climate and implications for productivity." Agricultural and Forest Meteorology 216 (2016): 177-187. Overall, the manuscript is easy to follow, and the approach may be evaluated in future studies. I think that the manuscript may be considered for publication after several minor edits. 1) more details about the calibrated threshold value using grounded phenological data are suggested. How many samples used for calibration and validation are unknown. How to determine the samples for calibration and validation is also required. 2) Figure 3, both x and y axis have the same titles. 3) Maps on the temporal and spatial changes of temperature in the study area may be needed.
Author Response
Reviewer 2
The manuscript aims to extract spatial and temporal variability of winter wheat phenology in Northern China. A calibrated threshold value of accumulated NDVI was proposed to extract different phenological events in winter wheat. Spatial and temporal variability of winter wheat phenology and theirs response to temperature were further assessed. In fact, extracting phenological metrics based on accumulated NDVI is not new, which can be found in the previous studies such as, Wu, Chaoyang, Xuehui Hou, Dailiang Peng, Alemu Gonsamo, and Shiguang Xu. "Land surface phenology of China's temperate ecosystems over 1999–2013: Spatial–temporal patterns, interaction effects, covariation with climate and implications for productivity." Agricultural and Forest Meteorology 216 (2016): 177-187. Overall, the manuscript is easy to follow, and the approach may be evaluated in future studies. I think that the manuscript may be considered for publication after several minor edits.
1) more details about the calibrated threshold value using grounded phenological data are suggested. How many samples used for calibration and validation are unknown. How to determine the samples for calibration and validation is also required.
Response: Thank you for your comments. The calibrated relative threshold was determined by multiyear average phenological date. More specifically, for each agro-meteorological station, the multiyear average phenological date was calculated from its ground phenological records of more than 20 years (i.e., 1993-2016). Then, the multiyear average phenological dates of all agro-meteorological stations across the study area (i.e., 378 stations) were used to regress with latitude, longitude, and altitude of corresponding stations to estimate multiyear average phenological date of each wheat pixel. On this basis, by applying the multiyear average phenological date to the multiyear average cumulative NDVI curve, the relative threshold for the phenological date of the specific wheat pixel could be determined following Eq. (1) and Eq. (2) in the manuscript. We have added these details to lines 211-213 and lines 221 in the revised manuscript.
On the other hand, even the relative threshold method has been widely used, but thresholds are mostly determined empirically and there is currently no suitable method to determine true value of relative thresholds for different phenological dates. Due to lack of true value of relative threshold, it is hard to validate the calibrated relative threshold directly. Therefore, we have not validated the calibrated threshold in the manuscript but validate the phenological date which was extracted base on the calibrated threshold as an alternative.
2) Figure 3, both x and y axis have the same titles.
Response: Thank you for pointing out typos of x and y axis in Figure 3. In fact, the y axis represents phenological dates extracted by CRTM method based on remotely sensed data. The “agro” in parentheses after the phenological name in y axis has been corrected with “RS”, which indicates that the phenological date was extracted from remotely sensed data.
3) Maps on the temporal and spatial changes of temperature in the study area may be needed.
Response: Thank you for your comments. Temperature is a fundamental parameter in this study. Adding maps on the temporal and spatial changes of temperature could help readers understand the manuscript better. But we focus on the relationship between temperature and phenological dates, not on temperature itself. In addition, there are ten temperature variables corresponding to different phenological dates and periods. Adding these maps could make the manuscript lengthy. Therefore, we did not add the maps after considering above reasons.

Reviewer 3 Report
The manuscript presents an innovative method for extracting phenological dates of winter wheat crops from remote measurements of their Normalized Difference Vegetation Index (NDVI).
Phenological events of wheat life (emergence, tillering, green-up onset etc.) are usually extracted from cumulative curve of NDVI observations at the crossing of given threshold. However, these thresholds are often estimated in approximate way and without accounting for their spatial variability.
Authors propose an innovative method for determining appropriately calibrated thresholds for each production zone. These thresholds are then used for studying both spatial and seasonal variation of phenological dates of winter wheat. Their correlations with seasonal temperatures are also discussed.
The subject is clearly explained. Maps are showing spatial distribution of phenological dates and their temporal trend. Results are widely discussed and comparison with previous studies are also made. Many bibliographic references are given for the interested readers. Conclusions are roughly supported by data, although some small correction could be necessary.
I have just two remarks to do. The first is about discussion of temporal trends and correlations. It looks that authors discuss trends and correlations considering both significant and not significant values. But if a trend (or a correlation coefficient) is not significant you cannot be sure it exists at all. Therefore, it would be more correct to drawn conclusion considering only significant values.
Just an example. At line 269 authors say that heading date show a temporal advancement in 73.64% of cases. However, according to Table 2, this value includes also non-significant trends. It would be more correct that it 42.52% of trends is advanced, 21.58% id delayed and 35.9% show no trend.
I recommend checking the discussion of results. Although it will not alter the general conclusions, it could bring some minor correction.
The second remark is about the flowchart of Figure 2a. It is so small that is difficult to read. I suggest putting it in a separated figure, increasing its size. It would be useful in helping the reader in better understand the complex calibration procedure.
For the above-mentioned reasons, I recommend a minor revision.
Author Response
Reviewer 3
The manuscript presents an innovative method for extracting phenological dates of winter wheat crops from remote measurements of their Normalized Difference Vegetation Index (NDVI). Phenological events of wheat life (emergence, tillering, green-up onset etc.) are usually extracted from cumulative curve of NDVI observations at the crossing of given threshold. However, these thresholds are often estimated in approximate way and without accounting for their spatial variability. Authors propose an innovative method for determining appropriately calibrated thresholds for each production zone. These thresholds are then used for studying both spatial and seasonal variation of phenological dates of winter wheat. Their correlations with seasonal temperatures are also discussed.
The subject is clearly explained. Maps are showing spatial distribution of phenological dates and their temporal trend. Results are widely discussed and comparison with previous studies are also made. Many bibliographic references are given for the interested readers. Conclusions are roughly supported by data, although some small correction could be necessary.
I have just two remarks to do. The first is about discussion of temporal trends and correlations. It looks that authors discuss trends and correlations considering both significant and not significant values. But if a trend (or a correlation coefficient) is not significant you cannot be sure it exists at all. Therefore, it would be more correct to drawn conclusion considering only significant values.
Just an example. At line 269 authors say that heading date show a temporal advancement in 73.64% of cases. However, according to Table 2, this value includes also non-significant trends. It would be more correct that it 42.52% of trends is advanced, 21.58% id delayed and 35.9% show no trend.
I recommend checking the discussion of results. Although it will not alter the general conclusions, it could bring some minor correction.
Response: Thank you for your suggestions. Following your advice, we have checked the results and discussion of the manuscript, and removed insignificant trends in describing the temporal trends, and drawn conclusions considering only significant values. (Page 17, lines 339-343; Page 19, lines 388-390; Page 28, lines 540-545)
The second remark is about the flowchart of Figure 2a. It is so small that is difficult to read. I suggest putting it in a separated figure, increasing its size. It would be useful in helping the reader in better understand the complex calibration procedure.
For the above-mentioned reasons, I recommend a minor revision.
Response: Thank you for your comments. We have increased the font size of Figure 2a and putted it in a separated figure (i.e., Figure 2 in the revised manuscript, lines 224-225). At the same time, we have also updated all the affected figure numbers accordingly.
